# Vascular Protective Effects of New Oral Anticoagulants in Patients with Atrial Fibrillation

**DOI:** 10.3390/jcm10194332

**Published:** 2021-09-23

**Authors:** Gyeong-Won Jang, Jung Myung Lee, Seung Woo Choi, Joan Kim, Young Shin Lee, Hyung Oh Kim, Hyemoon Chung, Jong Shin Woo, Jin Bae Kim, Woo-Shik Kim, Weon Kim

**Affiliations:** Division of Cardiology, Department of Internal Medicine, Kyung Hee University Hospital, Kyung Hee University, 26 Kyungheedae-ro, Dongdaemun-gu, Seoul 02447, Korea; sculptor1232@naver.com (G.-W.J.); cardioljm@khu.ac.kr (J.M.L.); swseungwoo1@gmail.com (S.W.C.); joankim6552@gmail.com (J.K.); conan2010@naver.com (Y.S.L.); hypnotica1999@hanmail.net (H.O.K.); bluesunny52@gmail.com (H.C.); snowball77@hanmail.net (J.S.W.); jinbbai@khu.ac.kr (J.B.K.); wskim1125@khu.ac.kr (W.-S.K.)

**Keywords:** atrial fibrillation, anticoagulants, atherosclerosis, cardiovascular diseases, endothelium

## Abstract

This study was designed to determine the efficacy of a new oral anticoagulant (NOAC) therapy for the prevention of endothelial dysfunction and atherosclerosis progression in patients with atrial fibrillation (AF). Sixty-five AF patients with a CHA2DS2-VASc score ≥2 without previous history of cardiovascular disease were registered and randomly assigned to either an NOAC group (dabigatran or rivaroxaban) or the warfarin group. Reactive hyperemia peripheral arterial tonometry (RH-PAT) measurements reflecting endothelial function were taken using Endo-PAT2000. Carotid intima–media thickness (IMT) was measured at baseline, 12 months, and 24 months, and several biomarkers were also analyzed. For the primary end point, the reactive hyperemia index (RHI) for the NOAC group was 1.5 ± 0.4 and that for the warfarin group was 1.6 ± 0.5. The left and right carotid IMT was 0.7 mm in the NOAC groups and 0.8 mm in the warfarin group. At 12 months, RHI was 1.6 ± 0.3 for the dabigatran group, 1.6 ± 0.5 for the rivaroxaban group, and 1.6 ± 0.3 for the warfarin group. The three groups did not differ statistically with respect to change in left and right carotid IMT at 12 and 24 months, respectively. The biomarkers for endothelial function and atherosclerosis were not significantly different. There was a trend of reduced P-selectin levels in the NOAC group compared to the warfarin group. In patients with AF, there were no significant differences in the prevention of endothelial dysfunction and atherosclerosis progression between the NOAC and warfarin groups.

## 1. Introduction

Activated coagulation factor Xa is known to play a central role in the coagulation cascade. Recent evidence further suggests that factor Xa has an important modulating effect in cellular signaling by the activation of protease-activated receptor (PAR) [1]. In fact, coagulation and inflammatory pathways interact with each other via factor Xa-mediated PAR activation on the arterial vessel wall and heart, and the resulting development of atherosclerosis and atrial fibrillation (AF) has been documented [2]. Preclinical studies have provided evidence for the effects of direct Xa or thrombin inhibition beyond anticoagulation, including anti-inflammatory and protective activities in atherosclerotic plaque development [1]. Evidence has demonstrated that direct thrombin inhibition impairs the formation and size of atherosclerotic plaques in addition to preventing progression of endothelial injury-associated stenosis in an apolipoprotein E-deficient mouse model [3,4]. However, the question remains as to whether these effects obtained in preclinical trials are similar in humans, for which there is scant evidence.

There are several non-invasive methods that allow researchers to assess endothelial function. Reactive hyperemia peripheral arterial tonometry (RH-PAT) and carotid IMT measurements are two such measurement tools [5], and previous studies suggest that the RH-PAT index is a useful predictor of coronary endothelial dysfunction [6,7]. Increased carotid IMT was associated with coronary artery severity [8,9,10].

This study aimed to determine the efficacy of new oral anticoagulant (NOAC) therapy for preventing endothelial dysfunction and atherosclerosis progression in AF patients. It was initially designed in 2015 and started enrollment in the same year, when NOAC and warfarin were equally classified as class I recommendations for stroke prevention in AF. However, new recommendations were published in the 2016 clinical guidelines, where the usage of NOAC was recommended over warfarin. Unfortunately, this made it more difficult to enroll patients, resulting in a break in the registration process. Patient enrollment was eventually terminated prematurely.

## 2. Materials and Methods

### 2.1. Study Subjects

We prospectively enrolled AF patients between 40 and 85 years of age and with CHA2DS2-VASc scores ≥ 2 (Table 1). The exclusion criteria were severe peripheral arterial disease (greater than Fontaine category IIb), grade 4 or higher cerebral infarction on the modified Rankin Scale, and proven coronary artery disease based on coronary angiogram. We also excluded patients with a range of concomitant comorbidities, including severe hepatic or renal dysfunction, uncontrolled congestive heart failure, hypertension, diabetes mellitus, hematological disorders, and allergy or hypersensitivity to the investigational drugs as well as pregnant or lactating women. Written informed consent was obtained from all patients, and the Ethics Review Board of Kyung Hee University Hospital approved this study. The two-year duration of the study period had a registration period from September 2015 to February 2016, with complete study duration from September 2017 to April 2018. For more information, please refer to our protocol article that has already been published [11].

### 2.2. Randomization

This study was a prospective, randomized, two-year follow-up study to further clarify the efficacy of NOAC in altering endothelial function and atherosclerosis progression in AF patients. The study design is shown in Figure 1. After enrollment, subjects were randomly assigned to the dabigatran group (110 or 150 mg twice/day; group 1), the rivaroxaban group (20 mg/day; group 2), or the warfarin group (controlled by international normalized ratio (INR) of 2–3; group 3). Clinical follow-up occurred at 1, 3, 12, and 24 months. Follow-up was conducted via telephone interviews or office visits. We assumed that NOACs would improve RHI values by an 8% difference, with no significant difference between the two NOACs. The expected difference in RHI was driven by clinical significance and previous medical treatments in other study populations [12,13,14]. To detect a statistically significant difference with a power of 80% with a two-sided α-level of 0.05, a sample of 165 patients (55 patients for each group) would be required. Assuming that the dropout rate would be 20%, the total sample size was set at 198 subjects [11]. However, due to the aforementioned reasons, it was difficult to enroll patients, and the registration speed slowed down due to the changes in the clinical guidelines and practice. Therefore, an interim analysis was conducted, and the results revealed no significant differences between the NOAC and warfarin groups, so fewer patients were enrolled.

### 2.3. Primary Outcome

The primary endpoint was defined as the change in the reactive hyperemia index (RHI) at 12 months. Secondary endpoints included changes in the right and left maximum IMT of the common carotid artery (CCA) and the internal carotid artery (ICA), mean IMT of the CCA and ICA at 12 and 24 months, 24-month cardiovascular events including cardiac death, stroke, acute myocardial infarction (AMI), overall cause of death, withdrawal of drug, or bleeding events.

### 2.4. RHI Measurement

Measurements were performed using a standard technique and device at baseline and at 12 months after randomization [15,16]. An RH-PAT 2000 device (Itamar Medical, Caesarea, Israel) was used for digital RH-PAT to evaluate endothelial function as previously described [5]. Impaired endothelial function was defined as logRHI <0.6, and favorable endothelial function was defined as logRHI ≥0.6.

### 2.5. Carotid IMT

CCA-IMT measurements were performed by experienced sonographers who were well trained in the use of 10-megahertz linear vascular probes (Vivid 7, GE Vingmed Ultrasound, Horten, Norway). Measurements were taken at baseline and 12 months after randomization. IMT was measured as the distance between two parallel echogenic lines corresponding to the blood–intima and media–adventitia interfaces on the posterior artery wall. Three IMT determinations were performed at the site of the thickest point with a maximum CCA-IMT and two adjacent points (1 cm upstream and 1 cm downstream from this site), and these three measurements were averaged (mean CCA-IMT). Carotid IMT was measured using dedicated software (Intimascope, Media Cross Co., Tokyo, Japan) by an examiner blinded to all clinical information.

### 2.6. Statistical Analyses

Demographic data were analyzed to identify pretreatment equivalencies and differences between the three study groups. Continuous variables are presented as the mean ± standard deviation and were compared using Student’s *t*-test or the Mann–Whitney test wherever appropriate. Non-normal distribution was identified from normality tests, and these data are presented as the median (interquartile range) and were compared using nonparametric methods (the Kruskal–Wallis or Wilcoxon signed-rank test). Categorical variables are presented as frequencies and percentages and were compared using the chi-squared or Fisher’s exact test wherever appropriate. Statistical significance was set at 0.05 (two-sided). All statistical analyses were performed using R software version 3.6.0. (R Foundation for Statistical Computing, Vienna, Austria).

### 2.7. Biomarkers of Atherosclerotic Plaque

Blood samples were obtained in serum separator tubes. After 20 min at room temperature, blood was centrifuged at 1000× *g* for 15 min. The serum was stored at −80 °C after dispensing 500 μL of serum into tubes. Concentrations of IL-6, TNF-α, p-selectin, and vWF were measured using the Quantikine and SimpleStep enzyme-linked immunosorbent assay (ELISA) kit according to the manufacturer’s protocol. A standard curve was created by reducing the data using computer software capable of generating a four-parameter logistic curve fit. The data were linearized by plotting the log of the concentration of the biomarkers versus the log of the optical density, and the best fit line could be determined by regression analysis.

## 3. Results

### 3.1. Patient Characteristics

From September 2015 to February 2016, 65 patients were randomly assigned. A total of 17 patients (26.2%) were excluded from the study due to loss at follow-up or side effects at 2-year follow-up (Figure 1). The patients’ characteristics are presented in Table 1. The average ages of participants in the dabigatran, rivaroxaban, and warfarin groups were 67.1 ± 9.4, 64.3 ± 7.7, and 67.7 ± 7.1 years, respectively. Males and non-smokers were more prevalent in all three groups, and more patients had hypertension than did not. There were no significant differences in patient characteristics between the three groups.

### 3.2. Reactive Hyperemia Index (RHI) and Carotid IMT

Table 2 presents the RHI and carotid IMT measurements at baseline, 12 months, and 24 months. At baseline, the dabigatran group’s RHI was 1.5 ± 0.4, the rivaroxaban RHI was 1.5 ± 0.4, and the warfarin RHI was 1.6 ± 0.5 (*p* = 0.487). The left and right carotid IMT was 0.7 mm in the NOAC groups and 0.8 mm (*p* = 0.697) in the warfarin group. At 12 months, the dabigatran group RHI was 1.6 ± 0.3, the rivaroxaban RHI was 1.6 ± 0.5, and the warfarin RHI was 1.6 ± 0.3 (*p* = 0.779, Figure 2). Carotid IMT values were not statistically different between the three groups. Notably, the three groups also did not differ statistically with respect to carotid IMT at 24 months.

### 3.3. Biomarkers of Atherosclerotic Plaque

Table 3 presents the endothelial and platelet activity biomarkers at baseline and 12 months. There was a trend of reduced p-selectin level in the NOAC groups compared to the warfarin group, but the results did not indicate statistically significant differences (Figure 3). The other biomarkers, namely IL-6, TNF-α, and vWF, showed no statistically significant differences between the NOAC and warfarin groups. There were no serious adverse events during this study.

## 4. Discussion

To the best of our knowledge, this is the first clinical study to investigate whether NOACs are indeed effective in altering endothelial function and atherosclerotic changes in patients with AF. Although AF may adversely affect endothelial function [15,17,18], it is not known whether vitamin K antagonists are helpful for such prevention. Previous studies showed that NOACs suppressed inflammatory cytokines and atherosclerotic cascades [3,4,19,20]. However, there was a lack of clinical data to corroborate the results. Interestingly, our trial showed that there was no difference in the ability to prevent endothelial dysfunction or atherosclerosis progression between the NOAC and warfarin groups. The results of this study provide new insights regarding our current understanding of NOAC mechanisms. The results should also help researchers to develop appropriate drug therapies for patients with AF and atherosclerosis. This will be much needed, as AF and related complications are becoming a significant health burden worldwide, with AF incidences rising at a staggering rate.

PARs are a family of G protein-coupled receptors which belong to four members (PAR-1 to -4). PAR-1 and PAR-2 are activated by factor Xa through a canonical G protein-dependent pathway in several cardiomyocytes and cardiac fibroblasts, which result in pathological cardiac remodeling in response to cardiac injury or stress [21]. In a mouse model, factor Xa inhibition was beneficial for prevention and regression of atherosclerosis, possibly mediated through reduced PAR activation [22]. Direct factor Xa inhibition was associated with slow progression of coronary atherosclerotic plaque compared with warfarin [23]. Experimental studies have suggested that a thrombin inhibitor improved endothelial function and decreased atherosclerosis in mice [24]. However, we found that there were no statistically significant differences between NOACs and warfarin with respect to vascular endothelial protection in high-risk patients with AF.

In a post hoc analysis of the X-VERT trial, rivaroxaban and warfarin showed similar influences on inflammatory biomarkers, including IL-6 [25]. Although the researchers found a similar reduction in inflammatory markers, we did not find a meaningful decrease in inflammation biomarkers in both NOAC and warfarin groups. It is possible that the results were different because of differences in the drugs that were used—the previous study only used rivaroxaban. Whereas, rivaroxaban and dabigatran were used in our study. However, in the RIVAL-AF trial, there were no significant differences in the changes in inflammatory cytokines such as IL-6 and TNF-α between the rivaroxaban and dabigatran groups. In a small observational study, a trend (*p* = 0.06) toward a reduction in P-selectin in the rivaroxaban treatment group compared to the control group was noted [26]. Our study also found a trend of reduced P-selectin level in NOAC-treated patients compared to warfarin-treated patients. The clinical significance of this finding is still unclear, thus further studies are needed. There are several other reasons why our results showed no significant differences. Firstly, previous animal studies administered higher doses of NOAC compared to the amount that was used in our study [22,24]. Secondly, our study was a small trial conducted over a short time period, and thus may not provide robust temporal statistical evidence.

Some potential limitations of this study should be considered. First, it included a relatively small number of patients due to early termination of the trial, with fewer patients enrolled than we had initially planned. Thus, our results should be interpreted with caution as the sample size may not be sufficient to detect significant differences between the warfarin and NOAC groups. Furthermore, the findings from this study cannot be generalized to all patients with AF. However, this issue is not unique to our study. Randomized controlled studies comparing the effects of NOACs and warfarin on human vascular endothelial cell function or biomarkers may suffer from the same ethical issues in the future. In this respect, this study provides invaluable data that will be difficult to obtain in future trials. Second, the primary outcome was not a hard endpoint for a clinical outcome but a surrogate marker that indirectly measures endothelial dysfunction. Therefore, future studies should directly explore clinical outcomes from vascular events that compare NOACs and warfarin.

## 5. Conclusions

There were no significant differences in the ability to prevent endothelial dysfunction and atherosclerosis progression between the NOAC and warfarin groups of AF patients.

## Figures and Tables

**Figure 1 jcm-10-04332-f001:**
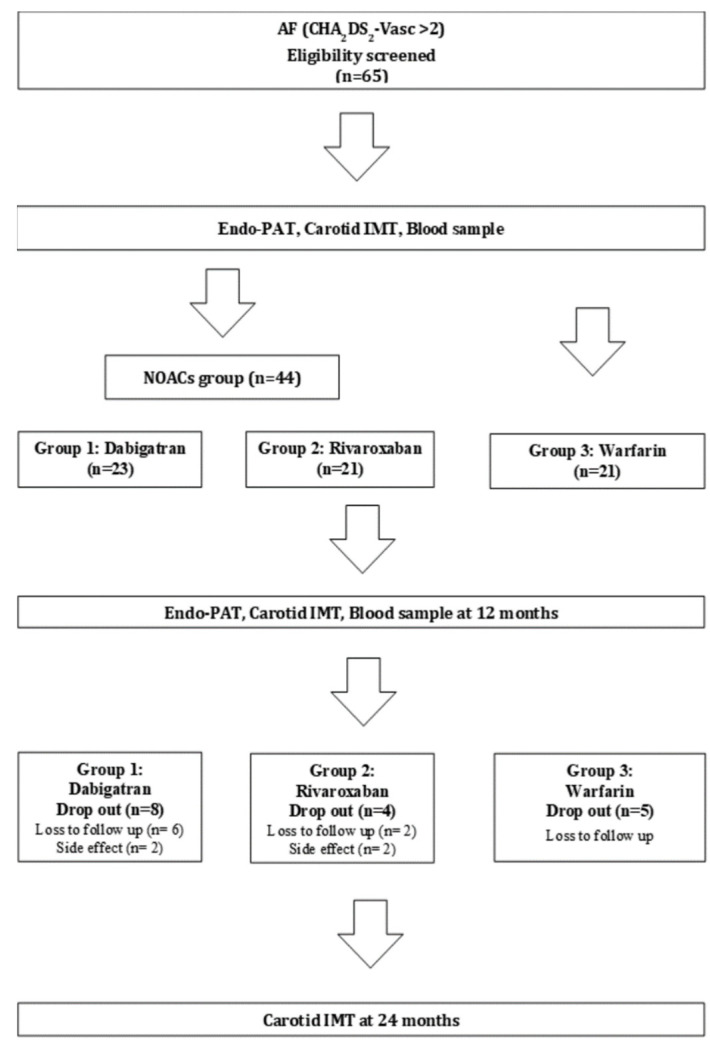
Flow diagram of study participants.

**Figure 2 jcm-10-04332-f002:**
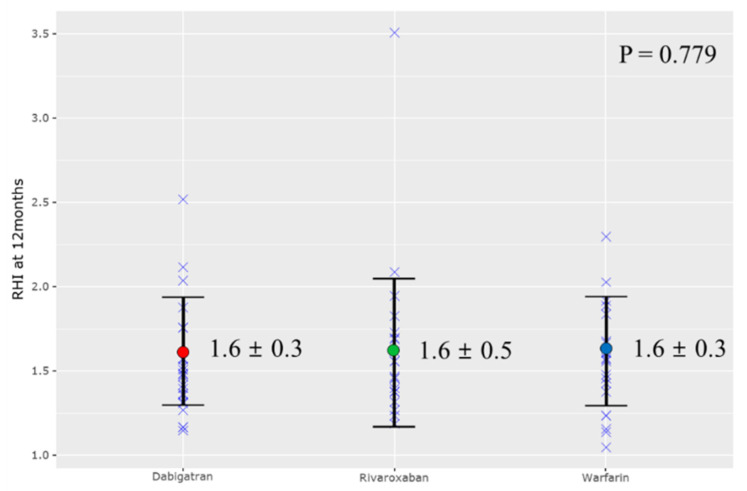
Reactive hyperemia index (RHI) at 12 months.

**Figure 3 jcm-10-04332-f003:**
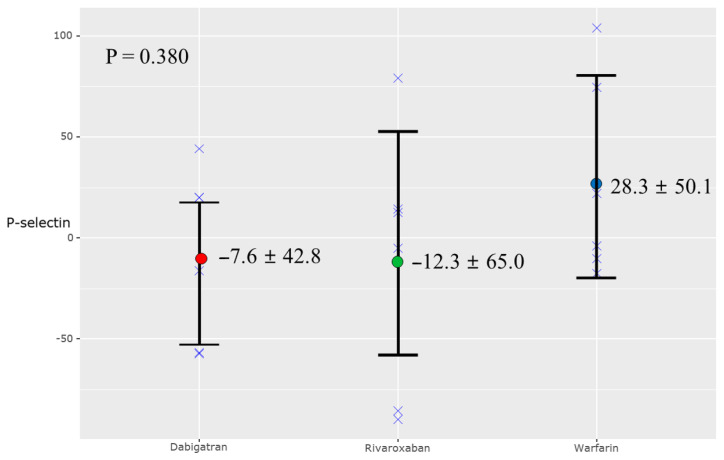
Change in P-selectin at 12 months.

**Table 1 jcm-10-04332-t001:** Baseline characteristics.

	Dabigatran	Rivaroxaban	Warfarin	*p*-Value
	(*n* = 23)	(*n* = 21)	(*n* = 21)	
Age (years)	67.1 ± 9.4	64.3 ± 7.7	67.7 ± 7.1	0.360
Sex				0.083
Female	11 (47.8%)	5 (23.8%)	4 (19.0%)	
Male	12 (52.2%)	16 (76.2%)	17 (81.0%)	
BMI (kg/m^2^)	25.3 ± 2.8	25.6 ± 3.1	25.3 ± 3.3	0.928
Smoking				0.586
Current	3 (13.0%)	3 (14.3%)	6 (28.6%)	
Former	5 (21.7%)	8 (38.1%)	5 (23.8%)	
Never	14 (60.9%)	10 (47.6%)	9 (42.9%)	
Medical history				
Congestive heart failure	6 (26.1%)	9 (42.9%)	6 (28.6%)	0.447
Diabetes mellitus	8 (34.8%)	2 (9.5%)	6 (28.6%)	0.133
Hypertension	17 (73.9%)	15 (71.4%)	11 (52.4%)	0.265
Dyslipidemia	12 (52.2%)	12 (57.1%)	8 (38.1%)	0.385
Current medication				
Aspirin	2 (8.7%)	4 (19.0%)	2 (9.5%)	0.519
ACEi or ARB	10 (43.5%)	8 (38.1%)	7 (33.3%)	0.787
Beta blocker	15 (65.2%)	12 (57.1%)	9 (42.9%)	0.323
Calcium channel blocker	8 (34.8%)	8 (38.1%)	9 (42.9%)	0.859
Statin	11 (47.8%)	11 (52.4%)	7 (33.3%)	0.430
Atorvastatin	4 (17.4%)	4 (28.6%)	4(14.3%)	0.475
Rosuvastatin	4(17.4%)	4((19.0%)	2(9.5%)	0.656

BMI = Body mass index; ACEi = Angiotensin-converting enzyme inhibitor; ARB = Angiotensin II type 1 receptor blocker. Data represent the number, frequency, or means ± SD.

**Table 2 jcm-10-04332-t002:** Reactive hyperemia index (RHI) and carotid IMT at baseline, 12 months, and 24 months.

	Dabigatran	Rivaroxaban	Warfarin	*p*-Value
	(*n* = 23)	(*n* = 21)	(*n* = 21)	
Baseline				
RHI	1.5 ± 0.4	1.5 ± 0.4	1.6 ± 0.5	0.487
Lt carotid IMT (mm)	0.7 ± 0.1	0.7 ± 0.1	0.8 ± 0.1	0.697
Rt carotid IMT (mm)	0.7 ± 0.1	0.7 ± 0.1	0.8 ± 0.2	0.495
Maximal plaque of Lt carotid IMT (mm)	1.9 ± 0.7	1.6 ± 0.5	2.0 ± 0.4	0.349
Maximal plaque of Rt carotid IMT (mm)	2.2 ± 0.7	1.9 ± 0.8	1.9 ± 0.6	0.452
12 months				
RHI	1.6 ± 0.3	1.6 ± 0.5	1.6 ± 0.3	0.779
Lt carotid IMT (mm)	0.8 ± 0.1	0.7 ± 0.1	0.8 ± 0.2	0.629
Rt carotid IMT (mm)	0.7 ± 0.1	0.7 ± 0.1	0.8 ± 0.1	0.145
Maximal plaque of Lt carotid IMT (mm)	1.8 ± 0.4	1.8 ± 0.5	1.6 ± 0.3	0.562
Maximal plaque of Rt carotid IMT (mm)	1.9 ± 0.7	1.6 ± 0.4	2.0 ± 0.4	0.218
24 months				
Lt carotid IMT (mm)	0.7 ± 0.1	0.7 ± 0.1	0.8 ± 0.1	0.901
Rt carotid IMT (mm)	0.7 ± 0.1	0.7 ± 0.1	0.7 ± 0.1	0.850
Maximal plaque of Lt carotid IMT (mm)	1.7 ± 0.3	1.9 ± 0.9	1.9 ± 0.7	0.714
Maximal plaque of Rt carotid IMT (mm)	1.9 ± 0.6	1.6 ± 0.5	2.0 ± 0.4	0.113

IMT = intima–media thickness; Lt = left; Rt = right.

**Table 3 jcm-10-04332-t003:** Biomarkers of atherosclerotic plaque at baseline and 12 months.

	NOACs	Warfarin	*p*-Value
Baseline			
IL-6 (pg/mL)	32.2 ± 8.9	28.3 ± 5.8	0.312
TNF-α (pg/mL)	1.8 ± 5.2	1.3 ± 3.7	0.800
P-selectin (ng/mL)	161.2 ± 51.8	166.0 ± 54.7	0.846
vWF (μg/mL)	6.7 ± 2.2	7.6 ± 2.6	0.416
12 months			
IL-6 (pg/mL)	35.6 ± 10.4	28.2 ± 5.0	0.094
TNF-α (pg/mL)	3.7 ± 5.1	2.8 ± 4.7	0.710
P-selectin (ng/mL)	155.8 ± 71.6	209.5 ± 94.3	0.161
vWF (μg/mL)	7.5 ± 2.9	6.8 ± 3.4	0.667
Change in biomarkers at 12 months			
IL-6 (pg/mL)	3.5 ± 11.4	−0.1 ± 1.6	0.279
TNF-α (pg/mL)	1.8 ± 5.6	1.5 ± 1.8	0.856
P-selectin (ng/mL)	−5.4 ± 54.7	43.5 ± 60.8	0.078
vWF (μg/mL)	0.7 ± 2.7	−0.8 ± 2.4	0.225

IL-6 = Interleukin-6; TNF-α = Tumor necrosis factor alpha; vWF = Von Willebrand Factor.

## Data Availability

All data are reported in the manuscript.

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
