# Peer review of "Vascular Protective Effects of New Oral Anticoagulants in Patients with Atrial Fibrillation"

_jcm, 2021, doi:10.3390/jcm10194332_

Round 1
Reviewer 1 Report
Major) DOAC has become the first line choice in most patients eligible for oral anticoagulants because previous studies have shown clinical benefit over warfarin. It is not clear whether RCT can be justified in order to test the impact of DOAC on endothelial function in such a situation. The authors did not show rigorous evidence that improving endothelial function can lead to better clinical outcomes. The authors should clarify this point in the introduction section. The order of methods is inappropriate. This section should be placed before the results section, otherwise, the manuscript is going to be confusing. I cannot find how to calculate the number of participants should be enrolled in this manuscript. If it was not calculated appropriately, the results can be really difficult to interpret. How much was the effect size expected to be? These information are critical. Minor) The authors concluded that there was no significant difference in the improvement of endothelial dysfunction, but this can be merely because of lack of power. This point should be emphasized. As the number of subjects in each group was relatively small, figures should show not only error bars but also each point to see the distribution.
Reviewer 2 Report
The study was performed well. The numbers in each group were small and there was a trend toward increased IL-6 in the warfarin group. I think a statement regarding the possibility of an underpowering would be appropriate.
Round 2
Reviewer 1 Report
The authors argued that Xa or thrombin inhibition potentially can have an anti-inflammatory effect beyond anti-coagulant, and the RH-PAT and carotid IMT are the indicators for reflecting the magnitude of endothelial function as the authors showed in the manuscript. However, even if preclinical studies showed the inflammatory markers are likely to be affected by Xa inhibition, that does not directly mean it is detectable because many factors may play role in levels of biomarkers and progression or regression of plaques.
Now, how much did they estimate the regression of atherosclerotic plaque would be? Did they estimate it to calculate the sample size? The authors showed that they estimated the sample size with a power of 80%, alpha-level of 0.05, and a dropout rate of 20%, but without the effect size. We can not calculate the sample size without the effect size, which is determined by lines of evidence based on previous studies or the authors' experimental or observational experiences.
Regardless of whether the study was carried out as it was initially planned, this part is critical to draw conclusions. Otherwise, they cannot say any of them because negative results help the others implement the next trials only if it is well-designed.
Therefore, I should say the conclusions in the manuscript are misleading. Even though they found no significant differences in endothelial function between the two groups, they should also state the reasons they speculated; the calculated effect size was not as large as they estimated, or they could not reach the goal of sample size, instead of just writing the results.
Author Response
English language editing is done and please see the attachment.
